# Molecular Alterations in Cutaneous Squamous Cell Carcinoma in Immunocompetent and Immunosuppressed Hosts—A Systematic Review

**DOI:** 10.3390/cancers15061832

**Published:** 2023-03-17

**Authors:** Denise Ann Tsang, Steve Y. C. Tam, Choon Chiat Oh

**Affiliations:** 1Department of Dermatology, Singapore General Hospital, Singapore 169608, Singapore; denise.tsang@mohh.com.sg; 2Education Resource Centre, Singapore General Hospital, Singapore 169608, Singapore; 3Duke-NUS Medical School, Singapore 169608, Singapore

**Keywords:** cutaneous squamous cell carcinoma, CSCC, molecular alterations, molecular profile, tumor microenvironment, immunosuppressed, immunocompetent, organ transplant

## Abstract

**Simple Summary:**

Cutaneous squamous cell carcinoma (cSCC) is the second most common form of skin cancer worldwide. Due to its high mutational burden, cSCC remains poorly understood at the molecular level. A considerable number of profiling studies have previously been performed on cSCC, but progress in the field has been slow due to the lack of consensus among these studies. Immunosuppressed patients (e.g. organ transplant recipients) are at greater risk of developing cSCC and this population experiences greater morbidity from this disease. In this study, we aim to review the molecular profile of cSCC among immunocompetent patients (ICPs) and immunosuppressed patients (ISPs) and to identify novel biomarkers of this disease. The molecular characterization of cSCC will shed new light on dysregulated pathways and potentially identify new key drivers of the disease, which may guide the direction of future targeted therapy in cSCC.

**Abstract:**

The characterization of cutaneous squamous cell carcinoma (cSCC) at the molecular level is lacking in the current literature due to the high mutational burden of this disease. Immunosuppressed patients afflicted with cSCC experience considerable morbidity and mortality. In this article, we review the molecular profile of cSCC among the immunosuppressed and immunocompetent populations at the genetic, epigenetic, transcriptomic, and proteometabolomic levels, as well as describing key differences in the tumor immune microenvironment between these two populations. We feature novel biomarkers from the recent literature which may serve as potential targets for therapy.

## 1. Introduction

Cutaneous squamous cell carcinoma (cSCC) is the second most common form of skin cancer worldwide. The global trends in cSCC are worrying. According to the Global Burden of Disease Study (GBD) 2017 database, which assessed worldwide trends in skin cancer across 195 countries from 1990 to 2017, the incidence of cSCC increased by 310% during this period. Statistically, cSCC ranks the highest among any cancer tracked by the GBD [1]. In Singapore, cSCC constitutes 28.3% of all skin cancers diagnosed, with the mean age of cancer diagnosis occurring at 72.7 years. The age-standardized incidence rates for cSCC across all ethnic groups locally ranged from 0.8 to 3.5 tumors per 100,000 person-years from 1968 to 2016 [2]. Although this figure is relatively low compared to those of Western countries, the rising incidence of cSCC worldwide poses a considerable public health threat.

Due to its high mutational burden, cSCC remains poorly understood at the molecular level. A considerable number of profiling studies have previously been performed on cSCC, but progress in the field has been hampered by the lack of consensus among these studies. Immunosuppressed patients are at greater risk of developing cSCC and this population experiences greater morbidity from the disease. The molecular characterization of cSCC sheds light on dysregulated pathways and identifies key drivers of the disease, thus heavily influencing the direction of future therapy in cSCC. In this study, we aimed to review the molecular profile of cSCC among immunocompetent patients (ICPs) and immunosuppressed patients (ISPs) and to identify novel biomarkers of this deadly disease.

## 2. Materials and Methods

The Preferred Reporting Items for Systematic Reviews and Meta-Analysis protocols (PRISMA-P) guided the methodology of this systematic review (Figure 1). A review protocol was entered into the PROSPERO database (Registry number: CRD42023394796). Online databases were queried for articles written in English from 1 January 2017 to 25 May 2022. The following databases were searched: PubMed, Medline, and Embase, using a combination of terms and their synonyms. Four primary search domains were employed, which were combined with the Boolean operator AND, whereas search terms contained within each domain were combined with the Boolean operator OR (Figure 2). The search was not restricted by study design.

The retrieved literature was screened by two independent authors (DAT, OCC) using titles and abstracts for inclusion. In situations in which the suitability of an article was uncertain, an assessment of the full text was carried out and discrepancies were resolved through a vote of consensus. Articles were selected based on the following inclusion criteria: (1) being written in the English language, (2) having an emphasis on cSCC, and (3) containing data focusing on the molecular profiling of cSCC. Articles were excluded for the following reasons: (1) not reporting original data, (2) having no data about molecular alterations, (3) having data which did not focus on cSCC, (4) the full text was not available, or (5) reporting on isolated cases. Human cell lines and representative animal models were not excluded. Additional articles found after the completion of the full-text review of the selected articles were added to our review if they met the eligibility criteria, did not meet the exclusion criteria, and were not duplicates.

## 3. Results

Our literature search enabled us to retrieve 1056 articles. An additional 16 articles were identified during the review and were also included. A total of 841 articles remained after duplicates were removed. After reviewing title and abstract, a further 703 articles were excluded. Full-text screening was performed on 138 articles, of which 80 were included for the final qualitative synthesis (Figure 1).

### 3.1. Molecular Alterations in Immunocompetent Hosts

A total of 68 articles discussed molecular alterations associated with cSCC in the immunocompetent population. Changes in genetic expression were reported in 22 articles and epigenetic alterations were described in six articles. A further 21 articles discussed transcriptomic changes in cSCC, whereas 15 studies examined proteometabolic expression. There were nine articles that analyzed the immune microenvironment of cSCC.

#### 3.1.1. Genetic Expression

Perineural invasion (PNI) is defined as tumor cells invading the perineural space. This feature has been established as one of the high-risk factors of cSCC, and its occurrence portends higher rates of local recurrence, metastases, and poor survival [3]. Three studies reported genetic alterations observed in cSCC with PNI (Table 1). All three articles examined cSCC of the head and neck (cSCCHN). Expression profiling of autophagy-related genes by Zheng et al. identified 239 differentially expressed genes (DEGs). These included upregulated genes in cSCC with PNI, such as *MAPK8*, which had the highest node degree (41), followed by *ERBB2* (31) and *HIF1A* (30), as well as downregulated genes such as TNF, with the highest node degree of 44, followed by *MYC* (42), *BCL2L1* (36), *MTOR* (34), and *PPARγ* (32). In addition, *RAB23* gene expression was positively correlated with *HIF1A* (*p* = 0.001, r = 0.690), *MAPK8* (*p* = 0.007, r = 0.583), and *ARFGAP1* (*p* = 0.000, r = 0.655) but negatively associated with *MTOR* (*p* = 0.002, r = −0.748) and *BCL2L1* (*p* = 0.015, r = −0.528) [4]. In another study, the top 10 DEGs identified in extensive PNI samples compared to combined focal and non-PNI samples included *PTGIS*, *THBS4*, *SRGN*, *FERMT2*, *NR4A3*, *TIMP1*, *SAMSN1*, *HGF*, *VCAN*, and *C3* [5]. Somatic missense mutations in *FGFR2* (40%) were exclusively seen in patients with histologic evidence of PNI in a study examining 10 cases of high-risk cSCCHN. Two novel mutations, *FGFR2 A380D* and *D528N*, were also observed in this cohort [6].

Genes involved in metastasis. Several genes were identified to play a role in metastatic cSCC (Table 2). In a recent study by Minaei et al., *PLAU*, *PLAUR*, *MMP1*, *MMP10*, *MMP13*, *ITGA5*, *VEGFA*, and various inflammatory cytokine genes were among the highest DEGs in metastatic compared to non-metastatic tumors and sun-exposed skin (SES) [7]. Additionally, the matrix metalloproteinase inhibitor genes *TIMP1* and *TIMP4* were also found to be differentially expressed in metastatic versus non-metastatic/SES tissues [7]. Yilmaz et al. conducted whole-exome and targeted sequencing of metastatic and localized cSCC and found increased *TP53* mutation frequencies in metastatic disease compared to localized disease (85% vs. 54%, respectively; *p* < 0.0001). The authors also found that the chromatin remodeling gene *KMT2D* had increased rates of mutation in the metastatic cSCC relative to non-metastatic tumors (62% vs. 31%) [8]. M.B. Lobl et al. performed targeted next-generation sequencing of matched, localized, and metastatic primary high-risk cSCC and found that the most frequently mutated genes in localized and metastatic cSCC, respectively, were *TP53* (70% vs. 70%), *CDKN2A* (20% vs. 40%), *KDR* (4% vs. 30%), *SMAD4* (30% vs. 20%), *NOTCH1* (20% vs. 10%), *PTEN* (10% vs. 20%), and *KIT* (10% vs. 20%). Notably, *HRAS* mutations were only observed in metastatic cSCC (20% of samples). In metastatic cSCC, the oncogenic cluster identified was *CDH1*, a gene responsible for making E-cadherin [9]. Patients with *TERT* promoter mutated cSCC were found to be at higher risk of local recurrence and lymph node metastases [10].

Genes with tumor-suppressive roles. Seven studies reported genes with tumor-suppressive roles in cSCC development (Table 3). Using immunocompetent mice, Alameda et al. showed that a moderate increase in *CYLD* expression levels reduced *NF-kB* activation, which favored the differentiation of tumor epidermal cells and inhibited its proliferation and decreased tumor angiogenesis [11]. In another study utilizing mouse models, the authors identified RIPK-PKP1 signaling as a novel axis involved in skin stratification and tumorigenesis, whereby the phosphorylation of PKP1’s N-terminal domain by *RIPK4* is necessary for epidermal differentiation [12]. Partial loss-of-function mutations in the genes encoding subunits of RNase H2 compromised ribonucleotide excision repair in mouse epidermis, which led to spontaneous DNA damage, a type I interferon response, skin inflammation, and cSCC development [13]. Using a murine skin carcinogenesis model, Sunkara et al. showed that *SFRP1* knockout resulted in the upregulation of genes involved in epithelial-to-mesenchymal transition, stemness, proliferation, and metastasis [14]. Moreover, Zhou et al. identified *HOXA9*, a direct target of onco-miR-365, to be significantly downregulated in human cSCC tumors and cell lines. The absence of *HOXA9* positively regulates HIF-1 and its downstream glycolytic regulators, which enhances glycolysis necessary for cSCC development, proliferation, migration, and invasion [15]. Using whole-genome sequencing, Thind et al. demonstrated significant recurrent copy number loss in the tumor suppressor genes *KANSL1* and *PTPRD* [16]. Likewise, *KMT2C*, *CREBBP*, and *NCOA2* were identified to demonstrate tumor-suppressive roles in the initiation and progression of human cSCC [17].

Genes with oncogenic roles. There were 27 unique genes which demonstrated oncogenic roles in the development of cSCC in our review (Table 4). The expression of *CDC258* and *CDC25C* was increased in cSCC compared to normal skin (*p* ≤ 0.001) and led to the suppression of apoptosis in cSCC cells by stimulating Pl3K/Akt signaling and survivin expression [18]. *FGFR2* activation and protein expression was increased in cSCC cells and found to be low in premalignant lesions and normal skin [19]. In a large study which sought to identify genetic variants in the HLA class II region associated with the risk of cSCC, the authors found that cSCC risk was associated with rs28535317 (OR = 1.20, *p* = 9.88 × 10^−11^), which corresponded to an amino-acid change from phenylalanine to leucine at codon 26 of *HLA-DRB1* (OR = 1.17, *p* = 2.48 × 10^−10^). An independent association was observed for a threonine-to-isoleucine change at codon 107 of *HLA-DQA1* (OR = 1.14, *p* = 2.34 × 10^−9^). Additional independent cSCC associations with *DQA1*05:01* and *DQA1*05:05* were also uncovered. Among the classical HLA alleles, cSCC was associated with *DRB1*01* (OR = 1.18, *p* = 5.86 × 10^−10^) [20]. The expression of *SERPINE1* was upregulated in all tumor cohorts compared to controls in a gene expression profiling study [7]. Thind et al. also observed that the 3′ UTR regions of *EVC* (48%), *PPP1R1A* (48%), and *LUM* (16%) were significantly functionally altered (Q-value < 0.05) in the non-coding genome of cSCC. Moreover, significant recurrent copy number gains in *CALR*, *CCND1*, and *FGF3* were observed for coding regions [16]. In a recent study by Yan et al., differential expression analysis demonstrated that many members belonging to the *S100* gene family, the *SPRR* gene family, and *FABP5* were significantly upregulated in cSCC cells. The top ten upregulated genes comprised of *S100A9*, *S100A8*, *SFN*, *S100A7*, *S100A2*, *SPRR2A*, *FABP5*, *ISG15*, *KRT6B*, and *KRT16*. Furthermore, seven keratin-encoded genes (*KRT5*, *KRT6A*, *KRT6B*, *KRT6C*, *KRT14*, *KRT16*, and *KRT17*), six genes from the *S100* family *(S100A2*, *S100A7*, *S100A7A*, *S100A8*, and *S100A9*), and five genes from the *SPRR* family (*SPRR2A*, *SPRR2B*, *SPRR2D*, *SPRR2F*, and *SPRR1B*) were significantly upregulated in cSCC cells [21].

Five new genes, *HEPHL1*, *FBN2*, *SULF1*, *SULF2*, and *TCN1*, were recently discovered in cSCC, which were significantly upregulated compared to normal skin (*p* < 0.001) and actinic keratosis (AK) (*p* < 0.01) [22]. Novel somatic mutations in *MLH1* (Q407*, Q426*, R423*) were also observed in an analysis of ten cases of high-risk cSCCHN [6]. In another study, *EPHA6* and *EPHA7* were identified as targets within the Eph-ephrin pathway, which is responsible for mitigating decreased cell viability in cSCC [23]. The authors also reported that *RAC1* was the largest hub within the Eph-ephrin signaling pathway (degree = 14). Novel mutations in BPI were identified in the same study (HGVS DNA reference: g.36938975G > A, g.36954682C > T) [23].

The expression of *METTL3* was observed to be upregulated in cSCC samples. Mechanistically, *METTL3*-mediated m^6^A modification regulated the expression of Δ*Np63* in cSCC and influenced its stem-like properties, including its colony-forming ability and tumorigenicity [24]. Quan et al. profiled CD133+ cSCC cells, observing the increased expression of multiple components of the *NOTCH* and *NF-κB* signaling pathways, including the key components *NOTCH1*, *IKKα* (*CHUK*), *RELA*, and *RELB*, which contributed to the maintenance of its stem-like phenotypic features [25].

**Table 4 cancers-15-01832-t004:** Genes with oncogenic roles.

Gene	Study Population	Results	Author	Year
*HLA-DQA1*	7238 cSCC cases and 56,961 controls	An independent association was observed for a threonine to isoleucine change at codon 107 of *HLA-DQA1* (OR = 1.14, *p* = 2.34 × 10^−9^). Independent cSCC associations with *DQA1*05:01* and *DQA1*05:05* were identified.	Wang et al. [20]	2018
*HLA-DRB1*		cSCC risk was associated with rs28535317 (OR = 1.20, *p* = 9.88 × 10^−11^) corresponding to an amino-acid change from phenylalanine to leucine at codon 26 of *HLA-DRB1* (OR = 1.17, *p* = 2.48 × 10^−10^). Among the classical HLA alleles, cSCC was associated with *DRB1*01* (OR = 1.18, *p* = 5.86 × 10^−10^).		
*CDC25B*	Mouse models, cultured human cSCC cell lines (SCC12B.2, SCC13, SRB1, SRB12, and Colo16)	Primarily cytoplasmic in skin and skin tumours. Increased in cSCC vs. normal skin (*p* ≤ 0.001)	Al-Matouq et al. [18]	2019
*CDC25C*		Primarily nuclear in the skin. Increased cytoplasmic signal in cSCC vs. normal skin (*p* ≤ 0.001)		
*FGFR2*	Human primary (SCC12A and SCC118) and metastatic cSCC cell lines (SCC7). Human normal skin samples (*n* = 9) AK (*n* = 9), cSCC (*n* = 28) and metastatic cSCC (*n* = 21)	Strong expression of *FGFR2* was observed in less than 5% of AK samples. In cSCC and metastatic cSCC, cytoplasmic and perinuclear *FGFR2* was noted in tumor cells in the invasive margin and expression was predominantly strong (55% and 60%, respectively)	Khandelwal et al. [19]	2019
*METTL3*	Cell lines: A431, HSC-1 Human cSCC samples from 8 patients	Expression of *METTL3* was significantly higher in the cSCC tissues. *METTL3* knock down decreased cell proliferation. Less number of colony formation in *METTL3* knock down groups vs. control (*p* < 0.05)	R. Zhou et al. [24]	2019
*NOTCH*	CD133+ cells	Inhibiting *NOTCH* reduced the CD133+ cell population (*p* < 0.05). *NOTCH* inhibition decreased DNA-binding activity of canonical NF-κB pathway subunit p65 (RelA), and non-canonical pathway subunits p52, and RelB (64%, 80% and 77% respectively [*p* < 0.001]) vs. control	Quan et al. [25]	2019
*BPI*	Human cSCC samples, uninvolved skin from 3 patients	Novel mutations in cSCC identified via the HitWalker2 prioritization analysis - HGVS DNA reference: g.36938975G > A, variant type: single AA change - HGVS DNA reference: g.36954682C > T, variant type: single AA change	Anderson et al. [23]	2020
*EPHA6*		Mitigated > 30% reductions in cell viability in cSCC		
*EPHA7*		Mitigated > 30% reductions in cell viability in cSCC Novel mutations in cSCC identified via the HitWalker2 prioritization analysis - HGVS DNA reference: g.93956676G > A, variant type: single AA change - HGVS DNA reference: g.93953241C > T, variant type: single AA change		
*RAC1*		*RAC1* was the largest hub within the Eph-ephrin signaling pathway (degree = 14)		
*FABP5*	6 human cSCC samples with matched adjacent skin samples, 3 healthy control skin tissues	Significantly overexpressed in cSCC tissues (*p* < 0.001). Decrease in cell proliferation measured at 48 h (*p* = 0.007), increase of cell apoptosis (*p* < 0.001) after *FABP5* knock down	Yan et al. [21]	2021
*S100A9*		Significantly overexpressed in cSCC tissues (*p* < 0.001). Ability of cell proliferation was significantly inhibited after 24h after *S100A9* knock down (*p* < 0.05)		
*FBN2*	Tissue samples of NNS (*n* = 6), NES (*n* = 6), AK (*n* = 6), and cSCC (*n* = 6)	Upregulated in cSCC vs. normal (*p* < 0.001) and AK (*p* < 0.01)	Zou et al. [22]	2021
*HEPHL1*		Upregulated in cSCC vs. normal (*p* < 0.01) and AK (*p* < 0.05)		
*SULF1*		Upregulated in cSCC vs. normal (*p* < 0.001) and AK (*p* < 0.01)		
*SULF2*		Upregulated in cSCC vs. normal (*p* < 0.001) and AK (*p* < 0.05)		
*TCN1*		Upregulated in cSCC vs. normal (*p* < 0.01) and AK (*p* < 0.05)		
*ZMIZ1*	Mouse models, human cSCC cell lines: A431(CRL–1555), SCC13, COLO16	*ZMIZ1* gene expression significantly increased within cSCC genomes (100-300× normal expression levels). High expression associated with poor outcomes in human cSCC patients, correlation threshold of 0.65 (Cox Proportional Hazards Regression, *p* = 0.0195; Log-rank Test, *p* = 0.028 at 50% quintile)	Aiderus et al. [17]	2021
*ZMIZ2*		*ZMIZ2* gene expression significantly increased within cSCC genomes (100-300× normal expression levels)		
*SERPINE1*	cSCCHN from 50 patients. 21 PRI-, 14 PRI+, 15 MET, matched SES	Upregulated in all tumor cohorts vs. SES (log2FC > 1, Padj < 0.05)	Minaei et al. [17]	2022
*CALR*	Matched tumor and blood DNA from 25 patients with regional metastases of cSCCHN	Gene amplification seen most commonly in tumor samples	Thind et al. [16]	2022
*CCND1*		Gene amplification seen most commonly in tumor samples		
*EVC*		3′ UTR region of *EVC* (48%) was significantly functionally altered in cSCC (Q-value < 0.05)		
*FGF3*		Gene amplification seen most commonly in tumor samples		
*LUM*		3′ UTR region of *LUM* (16%) was significantly functionally altered in cSCC (Q-value < 0.05)		
*PPP1R1A*		3′ UTR region of *PPP1R1A* (48%) was significantly functionally altered in cSCC (Q-value < 0.05)		

3′ UTR, three prime untranslated region; AK, actinic keratosis; *BPI*, bactericidal/permeability-increasing protein; *CALR*, calreticulin; *CCND1*, cyclin D1; CD133+ cells, purified cancer stem cell-like subset from primary cSCC tumors; *CDC25B*, cell division cycle 25B; *CDC25C*, cell division cycle 25C; *EPHA6*, EPH receptor A6; *EPHA7*, EPH receptor A7; EVC, Ellis van Creveld syndrome; *FABP5*, fatty acid binding protein 5; *FBN2*, fibrillin 2; *FGFR2*, fibroblast growth factor receptor 2; *HEPHL1*, hephaestin-like 1; HLA, human leukocyte antigen; *LUM*, lumican; *METTL3*, methyltransferase-like 3; NES, normal sun-exposed skin; NF-kB, nuclear factor-κB; NNS, normal non-sun-exposed skin; *PPP1R1A*, protein phosphatase 1 regulatory subunit 1A; *RAC1*, RAS-related C3 botulinum substrate 1; *SERPINE1*, serpin family E member 1; *SULF*, sulfatase; *TCN1*, transcobalamin I; *ZMIZ1*, zinc finger, MIZ-type containing 1; *ZMIZ2*, zinc finger, MIZ-type containing 2.

#### 3.1.2. Epigenetic Alterations

The epigenetic landscape plays a key role in the tumor microenvironment. Through epigenomic profiling, Latil et al. demonstrated that the priming of the epithelial–mesenchymal transition (EMT) occurs in the cancer cell of origin. The authors showed that chromatin opening was mainly associated with gene activation, whereas chromatin closing was associated with gene repression during EMT. The transcription factor (TF) motifs that were found to be upregulated with the highest statistical significance during EMT and that were enriched in the open chromatin regions of tumor mesenchymal-like cells (TMCs) were *Jun/AP1* (42%), *NF1* (45%), *Ets1* (10%), *bHLH* TFs (20–45%), *Nfatc* (27%), and *Smad2* (37%). Motif enrichment analysis of the chromatin regions that opened during tumorigenesis revealed a strong enrichment of the binding sites of TFs such as *Jun/AP1* (65%), *Ets1* (37%), *Runx* (29%), *Nf-kb* (22%), and *TEAD* (25%) [26]. Aiderus et al. defined two mutually exclusive paralogous oncogenic drivers, *Zmiz1* and *Zmiz2*, among the most recurrent drivers of cSCC development. Interestingly, all cSCC tumors with *Zmiz1/2* insertions had inactivating insertions in at least one gene involved in chromatin remodeling, suggesting that alterations in epigenetic regulation are essential in cSCC development [17]. Actin-like protein 6A (*ACTL6A*, *BAF53a*) is a key protein subunit of the SWI/SNF epigenetic chromatin regulatory complex. *ACTL6A* knockdown was reported to reduce cSCC cell proliferation, spheroid formation, invasion, and migration. The authors proposed that *ACTL6A* suppresses p21Cip1 promoter activity to reduce p21Cip1 protein as a mechanism for maintaining the aggressive epidermal squamous cell carcinoma phenotype [27].

DNA methylation is one of the various epigenetic mechanisms that control gene expression at the cellular level. The role of DNA methylation has been examined in three articles (Table 5). In a genome-wide DNA methylation profiling study, Hervás-Marín et al. found that invasive cSCC showed lower methylation levels than premalignant actinic keratosis. By contrast, high-risk non-metastatic and metastatic cSCC demonstrated higher methylation levels compared to low-risk cSCC (*p* < 0.001, two-sided *t*-test). Although overall, there was a risk-dependent change in DNA methylation patterns, mostly reflecting a gain of methylation, the authors proposed that a non-sequential and complex pattern of DNA methylation exists during cSCC progression [28]. Additionally, the expression of *ID4* and *UCHL1* was found to be significantly downregulated in cSCC tissues (*p* = 0.0111, *p* = 0.0205 respectively) and correlated with increased levels of promoter methylation (*p* = 0.00295, *p* = 0.0499, respectively) [29]. Hypermethylation of the *FILIP1L* locus was also observed in human cSCC and its expression was decreased in human cSCC cell lines [30].

#### 3.1.3. Transcriptomic Changes

##### MicroRNAs

MicroRNAs (miRNAs) are short noncoding RNAs that regulate the expression of protein-coding genes at the post-transcriptional level. We found seven studies that examined the expression of miRNAs in cSCC (Appendix A). Wimmer et al. demonstrated that the overexpression of miR-10b conferred the stem cell-characteristic of a capacity for 3D-spheroid formation to keratinocytes. Analysis of the downstream effects of miR-10b identified the actin- and tubulin cytoskeleton-associated protein DIAPH2 as a novel putative target of miR-10b [31]. Several other miRNAs were found to have oncogenic roles in cSCC development. MiR-21 and miR-205 were upregulated in invasive cSCC compared to cSCC in situ (*p* < 0.05) [32]. MiR-31 demonstrated increased expression in cSCC compared to normal skin cell lines (*p* < 0.01) and enhanced cSCC cell viability (*p* < 0.01) [33]. MiR-186 overexpression led to significantly enhanced cell proliferation, invasion, and migration in cSCC cells compared with controls (*p* < 0.01) [34]. Likewise, miR-221 also showed increased expression in cSCC compared to noncancerous tissues (*p* < 0.05) and promoted cell proliferation [35]. By contrast, two miRNAs, miR-130a and miR-181a, demonstrated tumor-suppressive effects in cSCC development. MiR-130a expression was almost undetectable in cSCC samples, and its overexpression led to a significant reduction in tumor volume (*p* < 0.05 at week 4 and *p* < 0.01 at week 5) [36]. Similarly, miR-181a showed low abundance in cSCC samples compared to normal skin (*p* = 0.0088). MiR-181a overexpressing cells were observed to grow slower and reach termination criteria at later time points (*p* = 0.0001) [37].

##### Circular RNAs

Circular RNAs (circRNAs) are single-stranded, covalently closed RNA molecules which serve biological functions through various mechanisms such as by acting as transcriptional regulators, miRNA sponges, and protein templates [38]. Several circRNAs were identified among studies which examined transcriptomic alterations in cSCC (Appendix A). Mahapatra et al. identified 55 circRNAs with significantly (*p* < 0.05) altered expression in cSCC. The majority of differentially expressed circRNAs (53 of 55) were downregulated and only two were upregulated in cSCC relative to healthy skin. Amongst the differentially expressed circRNAs, the most significantly downregulated circRNA was IFFO2, whereas the most upregulated was circ_EPSTI in terms of fold change [39]. In another study by Wei et al., 54 differentially expressed circRNAs were identified in cSCC. The top six upregulated circRNAs were hsa_circ_0068631, hsa_circ_0070-933, hsa_circ_0067772, hsa_circ_0003528, hsa_circ_0070934, and hsa_circ_0001955, and the top six downregulated circRNAs were hsa_circ_0022392, hsa_circ_0022383, hsa_circ_0005085, hsa_circ_0046449, hsa_circ_007-2279, and hsa_circ_0000375. Their parental genes were most enriched in the mitophagy, PPAR, HIF-1, and AMPK signaling pathways [40].

##### Transcription Factors

Transcription factors are involved in the conversion of genetic information from DNA into RNA. A large number of TFs with altered expression in cSCC were identified by Mahapatra et al. in a comprehensive analysis of transcriptomic changes in cSCC (Appendix A). FOXP3, ETS1, Oct-3/4, E2F1, and SOX2 were among the top 50 TFs with overrepresented binding sites among differentially expressed coding genes in cSCC [39]. Rose et al. examined the role of SMAD 2/3 in cSCC and found that lesional cSCC tissue exhibited significantly reduced activated SMAD2/3 compared to perilesional tissue (*p* < 0.001). High-risk tumor depths (≥4 mm) demonstrated a markedly significant negative dependence on both phosphorylated SMAD2 (C.C −0.214; *p* = 0.001) and SMAD3 (C.C −0.200; *p* = 0.002), which is consistent with a tumor-suppressive role for SMAD2/3 activators in cSCC [41].

##### Long Non-Coding RNAs

Long non-coding RNAs (lncRNAs) are RNAs longer than 200 nucleotides that are not translated into functional proteins. We identified 12 studies that examined the role of lncRNAs in cSCC development. In a recent comprehensive analysis of the lncRNA-mRNA co-expression network by Hu et al., a large number of lncRNAs were found to be differentially expressed between cSCC and healthy controls. The top seven differentially expressed lncRNAs are included in Appendix A. Three lncRNAs (PVT1, CTD-2521M24.9, and AL353997.3) were upregulated in cSCC and four lncRNAs (MIR4720, BX004987.5, CTD-2619J13.13, and LINC00478) were downregulated in cSCC. The same study also identified six previously unstudied lncRNAs (GXYLT1P3, LINC00348, LOC101928131, A-33-p3340852, A-21-p0003442, and LOC644838) which could contribute to cSCC progression [42]. LncRNA RP11-493L12.5 was found to be most upregulated (46.77-fold), whereas KB-1410C5.3/lnc-GRHL2 (0.005-fold) was the most downregulated among the significantly altered lncRNAs in a transcriptomic analysis of cSCC [39]. Zhang et al. proposed a novel c-MYC-assisted MALAT1-KTN1-EGFR axis, which contributes to cSCC progression [43]. Another lncRNA, AK144841, demonstrated 40-fold greater expression in cSCC than in healthy skin. Moreover, AK144841 was found to inhibit gene expression, specifically downregulating the expression of genes of the *Lce1* family, which is involved in epidermal terminal differentiation, and of anticancer genes (including *Cgref1*, *Brsk1*, *Basp1*, *Dusp5*, *Btg2*, *Anpep*, *Dhrs9*, *Stfa2*, *Tpm1*, *SerpinB2*, *Cpa4*, *Crct1*, *Cryab*, *Il24*, *Csf2*, and *Rgs16*), contributing to the dedifferentiation of tumor-forming keratinocytes, as well as molecular cascades involved in cSCC development [44]. LINC01048 was more greatly expressed in cSCC (*p* < 0.01) and recurrence tissues compared with adjacent normal and non-recurrence tissues. Mechanistically, LINC01048 was demonstrated to be transcriptionally activated by *USF1*, and the *USF1*-induced upregulation of LINC01048 promoted cell proliferation and apoptosis in cSCC by binding to *TAF15* to transcriptionally activate *YAP1* [45]. Li et al. reported the significant upregulation of LINC00319 in cSCC, which was associated with larger tumor size and lymphovascular invasion. Functional studies demonstrated that LINC00319 promoted CSCC cell proliferation, accelerated cell cycle progression, facilitated cell migration and invasion, and inhibited cell apoptosis. Mechanistic studies showed that LINC00319 exerts its oncogenic functions in CSCC through miR-1207-5p-mediated regulation of cyclin-dependent kinase 3 [46]. LncRNA EZR-AS1 expression was found to be significantly upregulated in cSCC tissues and cells compared with adjacent healthy tissues (*p* < 0.01), and its knockdown inhibited cSCC cell proliferation, migration, and invasion and promoted cell apoptosis [47]. LncRNA HCP5 exhibited the greatest upregulation in cSCC (logFC = 1.8) and the highest relative expression levels in cSCC compared to normal adjacent tissue (*p* < 0.001) in a recent study examining the competing endogenous RNA (ceRNA) network in cSCC [48]. The authors proposed that HCP5 may competitively bind to miR-138-5p to regulate EZH2 in cSCC cells, promoting autophagy and reducing apoptosis through the STAT3/VEGFR2 pathway.

By contrast, several other lncRNAs have demonstrated tumor-suppressive effects on cSCC development. Yu et al. proposed that lncRNA HOTAIR acted as a ceRNA to regulate PRAF2 expression through competitive binding to miR-326, contributing to cSCC development. HOTAIR overexpression significantly enhanced cSCC cell migration and proliferation and the EMT process, and its downregulation impeded these processes [49]. The expression of LINC00520, a novel lncRNA which has only been reported in a few tumors, was also reported to be downregulated in cSCC. The authors demonstrated that LINC00520 targeted EGFR, thus inhibiting the PI3K-AKT signaling pathway and suppressing cell proliferation and migration [50]. Other research showed that lncRNA TINCR overexpression promoted ALA-PDT-induced apoptosis and autophagy through the ERK1/2-SP3 pathway [51]. Finally, significant functional alterations were observed in the tumor-suppressing lncRNA LINC01003 (68% of specimens, Q-value: 0.0158) in a recent study analyzing cSCC samples from patients with regional metastases of the head and neck [16].

#### 3.1.4. Protein Expression

There were 45 proteins of interest identified among 14 studies. Six proteins identified from two studies were downregulated in cSCC (Appendix A). β-catenin expression was significantly reduced from normal in cSCC (93% to 69%, *p* < 0.001), and CK10 was found to be inversely correlated with cancer development (rs = −0.626, *p* < 0.001) [52]. Four other proteins, COL28A1, COL6A6, COL1A1, and TLN2, were also found to be decreased in cSCC. These proteins were mapped to two pathways (protein digestion and absorption and platelet activation), which were notably enriched in a study utilizing integrated proteomic and metabolomic analysis [53].

There were 25 unique proteins identified from seven studies which demonstrated an increased expression in cSCC (Appendix A). Sun et al. profiled the expression of a panel of protein biomarkers and found that CK17 (rs = 0.67, *p* < 0.001), CD44 (rs = 0.383, *p* < 0.001), EZR (rs = 0.717, *p* < 0.001), Hsp75 (rs = 0.593, *p* < 0.001), and Hsp90-α (rs = 0.660, *p* < 0.001) demonstrated positive correlations with cSCC development [52] In another study by Azimi et al., six proteins, APOA1, ALB, SERPINA1, HLA-B, HP, and TXNDC5, were differentially abundant in cSCC compared to AK [54]. In the same study, another five proteins, FLNA, IGHA1, MAP4, LGALS1, and FSCN1, were observed to be most frequently upregulated in cSCC samples relative to normal epidermis (adjusted *p* < 0.05; n > 8) [54]. Cox-2 expression was demonstrated to correlate with aggressive cSCC phenotypes. These tumors frequently showed mesenchymal-like spindle cell carcinomas with minimal keratinization [55]. Crawford et al. observed that TEM8 and CMG2 were significantly overexpressed in cSCC tissues compared to controls in a study utilizing cSCC cells originating from UV-irradiated mice [56]. LPCAT1 was also found to be upregulated in cSCC compared to primary human epidermal keratinocytes (*p* < 0.001), and its expression facilitated proliferation, impeded apoptosis, accelerated epithelial–mesenchymal transition, and enhanced cell metastasis in cSCC [57]. Z. Liu et al. showed that IGF2BP1 overexpression was necessary for cSCC cell growth by demonstrating that IGF2BP1 knockout decreased the levels of IGF2BP1-stablized mRNAs, including IGF2, CD44, Gli1, and Myc, which significantly inhibited cSCC cell survival and proliferation (*p* < 0.05) [58].

Several proteins were observed to contribute towards cSCC metastasis and differentiation (Appendix A). A. Azimi et al. identified 5037 proteins across primary and metastatic cSCC samples, of which 19 proteins, including ISG15, APOA1, and MARCKS, which have roles in metastasis, were increased and 11 proteins, including DMKN, APCS, and CST6, were decreased in metastatic cSCC lesions relative to the primary phenotypes (adj. *p*-value < 0.05) [59]. In another study, RAC1 was observed to be upregulated in metastatic compared to low-risk cSCC [60]. The same study also revealed that PABPC1, LGALS3BP, MARCKS, and SND1 were significantly increased in high-risk cSCC compared to low-risk cSCC [60]. Recently, uPAR was identified as a biomarker in metastatic cSCC in a study which demonstrated that uPAR protein levels were significantly increased in metastatic cSCC. The same study showed that increased expression of the uPAR protein was correlated with the downregulation of hsa-miR-340-5p and hsa-miR-377-3p [7]. Another protein, ENTPD1, was found to demonstrate significantly higher expression in human cSCC that metastasized compared to tumors that were nonmetastatic ([+] Met, *n* = 54, [−] Met, *n* = 51, *p* < 0.001) [61]. Finally, poorly differentiated cSCC displayed significantly higher cytoplasmic and lower nuclear iASPP expression compared to well-differentiated tumors [62]. ΔNp63 and TAp63 are isoforms encoded by the TP63 gene and play important roles in cSCC development. ∆Np63 was reported to prime the cancer cell of origin toward well-differentiated tumors [26]. In another study, homozygous deletion of TAp63 not only increased the susceptibility of mice to cSCC (46.67% in TAp63−/− mice vs. 20% in wild-type mice), it also enhanced metastasis [63].

#### 3.1.5. Metabolic Changes

The metabolic profile of cSCC was examined in one study (Appendix A). W. Chen et al. showed that several standard amino acids were mapped to regulatory pathways involved in cSCC development. The levels of L-glutamate, L-aspartate, L-arginine, L-glutamine, and L-phenylalanine were increased, whereas the level of arachidonate was decreased in cSCC samples compared to matched noncancerous tissue samples [53].

#### 3.1.6. Immune Landscape

Our review featured several studies that linked cSCC development with alterations within the innate immune system (Table 6). A predominant protumor gene expression signature of tumor-associated neutrophils (TANs) was identified in cSCC in a study utilizing a mouse cSCC cell line. The study demonstrated a significant increase in the fraction of neutrophils within cSCC compared to surrounding skin (*p* < 0.01), with TANs accounting for 30–80% of tumor-infiltrating CD45+ cells [64]. In addition, analysis of the expression of inflammasome components in cSCC cell lines and normal human epidermal keratinocytes (NHEKs) demonstrated the upregulation of the expression of AIM2 in cSCC cells (*p* < 0.01). The authors found that AIM2 knockdown caused the downregulation of many genes encoding proteins regulating the cell cycle, including CDK1, CDC7, CCNA1, CCNB3, KIF11, and TTK, which led to a reduction in cell viability and invasion, the suppression of growth, and the onset of apoptosis in cSCC cells [65].

The role of the complement system in cSCC development was examined in four studies utilizing human cSCC cell lines. C1r was demonstrated to upregulate the production of invasion-associated matrix metalloproteinases (MMPs) MMP1, MMP13, MMP10, and MMP12 and to promote the invasion of cSCC cells [66]. C3 and CFB mRNA expression was increased in cSCC compared to NHEK samples (*p* < 0.05) [67]. The role of C3 in cSCC development was highlighted in another study, which showed that C3a upregulated cyclin D1, cyclin E, VEGF, pro-MMP1, and pro-MMP2 expression. Moreover, the expression of stemness factors Sox-2, Nanog, Oct-4, c-Myc, and CD-44 was also observed to be stimulated by C3a [68]. CFI overexpression was shown to increase the production of MMP-13 and MMP-2, ERK1/2 activation, and cell proliferation and to enhance cSCC invasion (*p* < 0.01) [69].

The T-lymphocyte profiles of cSCC have been described in two articles (Table 6). In a study comparing T-lymphocyte profiles between keratoacanthomas (KA) and invasive human cSCC, CD8+ cells were found to be increased, whereas CD4+ cells were found to be decreased in cSCC relative to KA. In addition, the infiltration of FOXp3+ T-regulatory cells was lower in invasive cSCC compared to KA. Bauer et al. also demonstrated that PD1+ and PD-L1+ cells were enriched in cSCC and PD-L1 was correlated with the differentiation state in cSCC. For invasive cSCC, increased PD-L1 expression was correlated with enhanced infiltration of CD4+ and CD8+, as well as FOXp3+ T-cells (*p* = 0.0049, *p* = 0.0069, and *p* = 0.0397, respectively) [70]. Non-progressing primary cSCCHN samples were characterized by greater CD8+ (*p* = 0.006) and CD4+ (*p* = 0.004) T-cell responses, with numerically enhanced regulatory T-cells compared to tumors which metastasized [71].

**Table 6 cancers-15-01832-t006:** Immune alterations associated with cSCC development.

Immune Biomarker	Study Population	Findings	Author	Year
AIM2	Primary (*n* = 5) and metastatic (*n* = 3) human cSCC cell lines, NHEK (*n* = 5) Tissue samples: normal sun-protected skin (*n* = 15), AK (*n* = 71), cSCCIS (*n* = 60), and UV-induced cSCC (*n* = 81)	Elevated levels of AIM2 mRNA were noted in cSCC in vivo vs. normal skin (*p* < 0.01). A lower number of proliferating cells was observed in the xenografts established with cSCC cells transfected with AIM2 siRNA (21%) vs. control siRNA tumors (70%), *p* < 0.001.	Farshchian et al. [65]	2017
C1r	Human cSCC cell lines (UT-SCC-7, UTSCC- 12A, UT-SCC-59A, and UT-SCC-91)	*MMP1*, *MMP13*, *MMP10*, and *MMP12* were significantly downregulated after C1r knockdown (*p* < 0.001), offering evidence for the role of C1r in promoting the invasion of cSCC cells by increasing MMP production.	K Viiklepp et al. [66]	2022
C3	cSCC cell lines A431, Tca8113, SCC13, HSC-5 and HSC-1 and HaCaT	C3 mRNA expression was upregulated in all tumor cell lines and was more than 4.5 times higher in A431 and SCC13 cells.	Fan et al. [68]	2019
	Human cSCC cell lines (*n* = 8), NHEK (*n* = 11), mouse cSCC	Mean expression level of C3 mRNAs was higher in cSCC cells (*n* = 8), as compared to NHEKs (*n* = 11), *p* < 0.05. Growth of cSCC xenograft tumors with C3 knockdown was significantly reduced, as compared to control siRNA tumors, *p* < 0.05.	Riihilä et al. [67]	2017
CFB		Mean expression level of CFB mRNAs was higher in cSCC cells (*n* = 8), as compared to NHEKs (*n* = 11), *p* < 0.05. Migration rate of cSCC cells was significantly reduced after CFB knockdown, *p* < 0.01.		
CFI	Human cSCC cell lines (UT-SCC-1O5, UT-SCC-1O8, UT-SCC-7, and UT-SCC-59A)	Increase in the invasion of cSCC cells through 3D type I collagen (*p* < 0.001) and 3D Matrigel (*p* < 0.01) was noted following CFI overexpression.	Nezhad et al. [69]	2021
CD4+ T cells	Human cSCC from 31 patients: 9 NP, 22 DP. Within the DP group: 5 ISPs and 17 ACIS	Numbers of CD4+ T cells (*p* = 0.004) and FoxP3+ Tregs (*p* = 0.001) were higher in the NP group.	Ferguson et al. [71]	2022
	AK (*n* = 103), KA (*n* = 43), cSCC (*n* = 106)	Invasive cSCC showed less CD4+ cells vs. KA (*p* = 0.0158).	Bauer et al. [70]	2018
CD8+ T cells		Invasive cSCC demonstrated more infiltration of CD8+ cells vs. AK and KA (both *p* < 0.0001).		
	Human cSCC from 31 patients: 9 NP, 22 DP. Within the DP group: 5 ISPs and 17 ACIS	CD8+ T cells were greater in the NP group vs. the DP group (*p* = 0.006). CD8+ T cells were more proliferative (*p* *<* 0.0001), and expressed a greater (*p* < 0.0001) proportion of granzyme B in the NP group vs. DP group.	Ferguson et al. [71]	2022
FOXp3+ T cells	AK (*n* = 103), KA (*n* = 43), cSCC (*n* = 106)	Invasive cSCC showed less FOXp3+ T cells in the infiltrate vs. KA (*p* = 0.0063).	Bauer et al. [70]	2018
PD-L1		cSCC expressed significantly more PD-L1 in comparison with AK (*p* < 0.0001). PD-L1 expression was greater in moderately and poorly differentiated cSCC vs. well-differentiated cSCC (*p* = 0.0426).		
Podoplanin	Mouse cSCC, human cSCC cell lines	Podoplanin interacts with both CD44s and CD44v (CD44v3-10, CD44v6-10, and CD44v8-10) isoforms expressed in SCC cell lines.	L Montero-Montero et al. [72]	2020
Tumor-Associated Neutrophils	Mouse models, mouse cSCC cell line: mSCC38	Significant increase in the proportion of neutrophils within cSCC vs. surrounding skin (*p* < 0.01), with TANs accounting for 30–80% of tumor-infiltrating CD45+ cells. TANs contribute to cSCC development by limiting effector CD8+ T cell responses.	Khou et al. [64]	2020

AK, actinic keratosis; C1r, complement C1r; C3, complement factor 3; CFB, complement factor B; CFI, complement factor I; cSCCIS, cSCC in situ; DP, progression to metastases; KA, keratoacanthoma; *MMP*, matrix metalloproteinase; NHEK, normal human epidermal keratinocyte; NP, non-progressors; siRNA, small interfering RNA.

### 3.2. Molecular Alterations in Immunosuppressed Hosts

There were 14 studies which focused on the immunosuppressed population. Among these studies, the majority focused on the general organ-transplant population, including renal transplant recipients (RTRs). A small number of articles studied populations with unspecified immunosuppression.

#### 3.2.1. Tumor Immune Microenvironment

Six studies examined the tumor immune microenvironment of cSCC in immunosuppressed individuals (Table 7). Sun et al. reported that CD3-IL-17+ cells accounted for more than 90% of the total IL-17A-producing cells in cSCC tissue from ISPs, whereas the ratio of CD3+IL-17+ versus CD3-IL-17+ cells in ICPs varied [73]. Transplant-associated SCC (TSCC) patients showed decreased CD8+ T effector cells (*p* < 0.05) and demonstrated an increased frequency (>6 fold) of primary cSCC [74]. B7-H3, a known immune checkpoint molecule and oncogene, was widely expressed in tumors from ICPs compared to ISPs (*p* = 0.025) [75]. In a study examining organ transplant recipients (OTRs) with cSCC, the expression of PGE_2_ (OR = 1.9, 95% CI = 1.1–3.4, *p* = 0.002), POMC (OR = 1.5, 95% CI = 0.99–2.0, *p* = 0.05), and TNF-α (adjusted OR = 1.4, 95% CI = 0.99–2.0, *p* = 0.05) was associated with tender lesions [76]. Farshchian et al. reported that AIM2 expression was significantly more abundant in cSCC (*n* = 57) compared with cSCC in situ (cSCCIS) in OTR-derived tissues (*n* = 59, *p* < 0.001) [65]. Using high-dimensional and spatial analysis, Ferguson et al. observed that immune checkpoint receptors, including PD-L1, PD-L2, IDO, and TIM3, were upregulated in metastatic cSCC amongst ICPs, but this increased expression was lacking amongst ISPs [71].

#### 3.2.2. Gene Polymorphisms

Seven unique alleles were identified across three studies (Table 8). The IRF4 rs12203592 T allele was associated with a significantly increased hazard ratio for the time to first cSCC (HR = 1.36, *p* = 0.02) using univariate analysis. This association was maintained when adjusted for age, gender, organ transplanted, and Fitzpatrick skin type (HR = 1.34, *p* = 0.04) [77]. By contrast, OTRs homozygous for the brown eye alleles rs916977 (GG) and rs12913832 (AA) exhibited significant delays in the time to first cSCC post-transplant relative to OTRs homozygous for blue eye alleles (HR = 0.34, *p* < 0.001 and HR = 0.54, *p* = 0.012, respectively) [78]. Likewise, the *SLC45A2* rs16891982 C allele was associated with a decreased hazard for cSCC in univariate analysis (HR = 0.58, *p* = 0.04). This effect was comparable but not significant with the application of a multivariate model (HR = 0.74, *p* = 0.06) [77]. In a later study by Kuzmanov et al., a genome-wide association study (GWAS) identified the SNV rs34567942 to be significantly associated with cSCC in OTRs (*p*-value threshold of 5 × 10^−8^) [79].

#### 3.2.3. Genetic and Epigenetic Alterations

Peters et al. identified 16 differentially methylated regions in RTRs, including *ZNF577* and *FLOT1* [80] (Table 9). A subsequent study revealed that higher DNA methylation of *SERPINB9* occurred in RTRs who developed cSCC than in those who did not. The median DNA methylation of *SERPINB9* was 58.7% (range 32.5–81.3%) for region 1 and 54.4% (30.0–78.5%) for region 2 in patients with cSCC, and 50.2% (21.8–77.5%) for region 1 and 46.4% (22.1–74.0%) for region 2 in the non-cSCC patients (region 1: *p* = 0.004; region 2: *p* = 0.008) [81].

In another study examining the tissue and circulating expression of miRNAs in OTRs with and without cSCC, the authors found that mir-1246 and mir-1290 were associated with cSCC in OTRs (*p* = 0.013 and *p* = 0.037, respectively) [82] (Table 9). In addition, a previously unknown mutational signature, termed signature 32, was discovered in cSCC samples of OTRs during whole-exome sequencing and mutational signature analysis. Signature 32 describes predominately C > T mutations (75%) in combination with C > A, T > A, and T > C mutations. Its pattern was observed to be putatively associated with azathioprine treatment and distinct from any of the previously known mutational signatures [83]. An analysis of treatment times revealed a strong positive correlation with the estimated time of azathioprine exposure and the prevalence of signature 32 (Spearman’s rank order correlation rs (26) = 0.679, *p* < 0.0001) [83].

**Table 9 cancers-15-01832-t009:** Genetic and epigenetic alterations associated with cSCC in the immunosuppressed.

Gene/RNA of Interest	Study Population	Results	Author	Year
*FLOT1*	27 RTRs with SCC and 27 RTRs without SCC	Hypomethylated in patients with cSCC.	Peters et al. [80]	2018
*ZNF577*		Hypermethylated in the group with de novo cSCC after transplantation.		
Signature 32	40 cSCC samples from 37 patients; ISPs (*n* = 30), ICPs (*n* = 7)	Strong positive correlation with the estimated time of azathioprine exposure and the prevalence of signature 32 (Spearman’s rank order correlation r_s_(26) = 0.679, *p* < 0.0001). Most SMG gene mutations observed (including *NOTCH1/2*, *TP53*, and *CDKN2A*) were attributed to azathioprine signature 32 (66.2%).	Inman et al. [83]	2018
*SERPINB9*	Cohort 1: 19 RTRs with cSCC and 19 RTRs without cSCC Cohort 2: 45 RTRs with cSCC and 37 RTRs without cSCC	Higher DNA methylation of *SERPINB9* in RTRs who developed cSCC vs. those who did not. Median DNA methylation of *SERPINB9* was 58.7% (range: 32.5–81.3%) for region 1 and 54.4% (30.0–78.5%) for region 2 in patients with cSCC and 50.2% (21.8–77.5%) for region 1 and 46.4% (22.1–74.0%) for region 2 in the non-cSCC patients (region 1: *p* = 0.004 and region 2: *p* = 0.008).	Peters et al. [81]	2019
*HLA-DRB1*13*	46 RTRs who developed cSCC after transplant	HLA-DRB1*13 was associated with risk of cSCC in RTRs after transplant (HR = 2.24, 95% CI = 1.12–4.49, *p* = 0.023).	Kim et al. [84]	2020
mir-1246	8 OTRs with cSCC, 8 OTRs without cSCC	mir-1246 was significantly upregulated in both tumor tissue and serum in OTRs with cSCC vs. those without (*p* = 0.013).	Geusau et al. [82]	2020
mir-1290		mir-1290 was significantly upregulated in both tumor tissue and serum in OTRs with cSCC vs. those without (*p* = 0.037).		

*FLOT1*, flotillin-1; HLA, human leukocyte antigen; HR, hazard ratio; OTRs, organ transplant recipients; RTRs, renal transplant recipients; *ZNF577*, zinc finger protein 577.

## 4. Discussion

### 4.1. Novel Molecular Targets in CSCC

#### 4.1.1. Genomic Biomarkers

Substantial work has been carried out in ICP populations, with noteworthy findings including the major roles of *TP53*, *NOTCH*, *TGFβ*, and *CDKN2A* in the development of cSCC in this population [85,86,87,88]. In the recent literature, novel genetic alterations in cSCC have emerged, with roles in autophagy, perineural invasion, and metastasis.

The dysregulation of autophagy contributes towards cancer development, and recent research has shown that autophagy might play a pivotal role in the pathogenesis of cSCC [89,90]. Zheng et al. identified several key autophagy-related DEGs, namely, *HIF1A*, *MAPK8*, *mTOR*, *BCL2L1*, and *RAB23*, which were involved in cSCCHN with clinical PNI [4]. A close correlation may exist between autophagy and cSCC outcomes, and autophagy-related genes are a promising treatment target for skin cancer.

Moreover, a number of genes, including *PLAU*, *PLAUR*, *MMP1*, *MMP10*, *MMP13*, *ITGA5*, and *VEGFA*, were recently revealed to be differentially upregulated in metastatic compared to non-metastatic cSCC [7]. These genes are involved in cellular pathways and functions that support matrix remodeling, epithelial-to-mesenchymal transition, cell survival, and migration, all of which play important roles in tumor metastasis [7]. The discovery of these genes could be useful in identifying primary cSCC tumors with metastatic potential. TIMP4 was previously reported to be downregulated in non-cutaneous head and neck SCCs [91], but its downregulation in metastatic cSCC compared to sun-exposed skin was only recently uncovered [7].

Novel somatic mutations in *MLH1* (Q407*, Q426*, R423*) were observed in an analysis of 10 cases of high-risk cSCCHN [6]. These mutations led to premature truncation and loss of the C-terminal dimerization domain. Two other new mutations, *FGFR2 A380D* and *D528N*, were discovered in a cohort of patients with high-risk cSCCHN. These mutations resulted in changes within the transmembrane domain and the protein tyrosine kinase domain, respectively, and were exclusively seen in patients with histologically proven PNI. Five novel genes, *HEPHL1*, *FBN2*, *SULF1*, *SULF2*, and *TCN1*, were also observed to be significantly upregulated in cSCC compared to normal skin [22]. In a human-to-mouse comparison of cSCC tumors, the authors discovered that miR-30c-2* and miR-497 were under-expressed in TAp63-deficient cSCC. In the same study, a seven-gene signature was identified, including five putative targets of miR-30c-2* (*FAT2*, *ITGA6*, *KIF18B*, *ORC1*, and *PKMYT1*) and four predicted targets of miR-497 (*AURKA*, *CDK6*, *KIF18B*, and *PKMYT1*), which were frequently overexpressed in cSCC [63]. Further studies on the roles of these genes in cSCC development could further enhance diagnosis and risk stratification and broaden therapeutic options.

#### 4.1.2. Transcriptomic Biomarkers

Accumulating evidence demonstrates that the dysregulation of miRNAs plays an essential role in cSCC development and progression. Our review showed that miR-10b31, miR-21 [32], miR-31 [33], miR-186 [34], miR-205 [32], and miR-221 [35] were upregulated, whereas miR-130a [36] and miR-181a [37] were downregulated in cSCC. Interestingly, miR-21 and miR-205 were induced in invasive cSCC compared to cSCCIS (*p* ≤ 0.05) [32] and these may serve as useful biomarkers for the risk stratification of cSCC.

Four previously unstudied circRNAs, circ_IFFO2, circ_TNFRSF21, circ_KRT1, and circ_POF1B, which were identified by Mahapatra et al., showed differential expression in cSCC [39]. CircRNAs have recently emerged as a novel member of the noncoding cancer genome, with roles in controlling cancer gene expression through mechanisms such as decoys to sponge miRNAs and as regulators of transcription and alternative splicing [92]. The role of circRNAs in cSCC is being studied at an increasing pace, and these molecular biomarkers may introduce new therapeutic opportunities for cSCC in the foreseeable future.

LncRNAs have also garnered interest as novel regulators of gene expression in the recent literature. Hu et al. identified six previously unstudied lncRNAs, including GXYLT1P3, LINC00348, LOC101928131, A-33-p3340852, A-21-p0003442, and LOC644838 [42]. These lncRNAs demonstrated co-expression with the mRNAs ACY3, NR1D1, and MZB1, which could contribute considerably to cSCC progression by regulating apoptosis induced by endoplasmic reticulum stress, cellular signal transduction, and autophagy [42]. MALAT, another lncRNA, was highlighted as part of a novel c-MYC-assisted MALAT1-KTN1-EGFR axis responsible for cSCC progression [43]. Transcriptomic sequencing identified KTN1 as the key mediator through which MALAT1 positively regulated the expression of EGFR and contributed to cSCC development [43].

#### 4.1.3. Proteomic Biomarkers

Ten proteins identified by Sun et al. (CK10, CK17, CD44, EZR, E-cadherin, b-catenin, Hsp75, Hs-p90-α, EXOSC10, and SOD2) showed disease-progression-specific significance in cSCC [52]. The chaperone proteins, Hsp75, Hsp90-α, and SOD2, were the only three proteins that demonstrated a correlation with differentiation stages of cSCC. The expression of chaperone proteins increased in parallel at each stage of cSCC development. In contrast, SOD2 had no effect on early skin damage, as its expression was comparable in normal and pre-cancerous samples [52]. Increased Hsp promotes cancer progression by participating in microenvironment conditioning [93], which makes it an appealing target in cSCC treatment. Another protein reported to demonstrate an association with cSCC differentiation status was iASPP [62]. This protein acts by inhibiting the p53 and NF-kB signaling pathways during cSCC development. The authors also described a previously unreported mechanism in cSCC by which increased iASPP levels bind and repress MITF expression, reducing TRPM1 and consequently miR-211 expression, which contributes to the increased stabilization of p63 observed in cSCC [62]. Epigenetic dysregulation of this autoregulatory feedback loop which promotes cSCC development could serve as a novel candidate for targeted therapy.

Proteins associated with metastasis may be useful in discriminating between primary and metastatic cSCC lesions. Twelve protein markers were identified by Azimi et al. that could be used for the early diagnosis and risk stratification of primary cSCC lesions. Of these, six proteins including ISG15, APOA1, MARCKS, EFHD2, STMN1, and ACBD3 were increased and six proteins including DMKN, APCS, CPA3, KRT79, CST6, and CMA1 were decreased in metastatic cSCC compared to the primary lesions [59]. The exact roles of most of these proteins have not been well established, although proteins with increased abundance in metastatic cSCC have been reported to increase tumor invasion and metastasis in other cancers. Another six proteins, APOA1, ALB, SERPINA1, HLA-B, HP, and TXNDC5, were differentially abundant in cSCC compared to AK [54]. These proteins are useful biomarkers for distinguishing cSCC from its precursors and may serve as potential molecular targets for selective treatment of these tumors.

#### 4.1.4. Immune Biomarkers

Defining the immune phenotypes within the tumor microenvironment (TME) of cSCC has been challenging, and immunoregulatory mechanisms leading to cSCC development remain unclear. T-lymphocyte profiles differ between AK, KA, and invasive cSCC [70]. Ferguson et al. showed that effective responses from both CD8+ and CD4+ T-cells in the TME are necessary for immune control of primary cSCCHN [71]. That study provided insights into the early events in cSCC that dictate the immune responses in primary tumors and influence disease outcomes. The ability to predict metastatic potential at the time of diagnosis of primary cSCC could be used to offer personalized care, including disease surveillance strategies and recognizing patients who will benefit the most from adjuvant therapy.

The complement system is a vital part of innate immunity against pathogen invasion. A number of complement factors have been implicated in solid organ cancers. In cSCC, complement factor H expression has previously been reported to enhance the growth and migration of cSCC cell lines [94]. Although C3 has been shown to be highly expressed in cSCC, its role remains unclear. Fan et al. elucidated a novel correlation between complement anaphylatoxin C3a and cSCC stemness, which could provide insights into the role of C3 in cSCC tumorigenesis [68]. CFB [67], CFI [69], and most recently C1r [66] were also reported to contribute towards cSCC progression.

Khou et al. discovered a predominant protumor gene expression signature of TANs in cSCC compared to normal skin. The authors found that in vivo depletion of neutrophils impeded tumor growth and significantly enhanced the frequency of proliferating IFN-gamma-producing CD8+ T-cells [64]. Mechanisms that limited antitumor responses involved high arginase activity, the production of reactive oxygen species (ROS) and nitrite (NO), and the expression of programmed death-ligand 1 (PD-L1) on TANs, concomitantly with an induction of PD-1 on CD8+ T-cells, which was correlated with tumor size [64]. These findings raise the possibility of targeting neutrophils and PD-L1-PD-1 interaction in the treatment of cSCC.

### 4.2. Key Molecular Alterations in the Immunosuppressed

CSCC is the most common post-transplant malignancy in organ-transplant recipients (OTRs) [95]. OTRs are 65–100 times more likely to develop cSCC than the general population [96,97]. Transplant patients with cSCC experience considerable mortality of 4.94 per 100,000 person years [98] and a greater morbidity ratio of up to 250 times compared with ICPs [99]. These differences are thought to arise from a more aggressive phenotype, a higher probability of metastasis (approximately 7%), and a greater recurrence rate (7–45%) [100,101,102].

#### 4.2.1. DNA Methylation—A Novel Risk Factor for CSCC Development

Epigenetic alterations, in the form of DNA hypomethylation and hypermethylation, have previously been associated with the development of skin cancer in ICPs [103,104], although less is known regarding its role in ISP. A comparison of RTRs who developed a future de novo post-transplant cSCC and those who did not identified 16 differentially methylated regions. Noteworthy genes included *ZNF577* and *FLOT1*, which encode a zinc-finger protein and a protein involved in T-cell migration, respectively [80]. The high DNA methylation of *SERPINB9* in circulating T-cells was demonstrated in RTRs before the clinical onset of cSCC, as well as during recurrent post-transplant cSCC [81]. The DNA methylation of these genes could serve as a novel risk factor for the development of de-novo and subsequent post-transplant cSCC and provide opportunities for pretransplant risk stratification and post-transplant surveillance of cSCC.

#### 4.2.2. Genetic Polymorphisms May Confer Protection against CSCC Development

Previously, GWAS performed for cSCC revealed that an increased cSCC risk was associated with polymorphisms in six pigment-related loci and that these influenced the cSCC risk independently of pigment-related phenotypes [105]. Moreover, variants at the *HERC2/OCA2* locus have been associated with pigmentation phenotypes and the risk of developing several types of skin cancer. Candidate genes and GWAS have linked blue-eye-associated alleles at the *HERC2/OCA2* locus with an increased risk of cSCC in ICPs, but the roles of these alleles in ISPs are less established [78].

Wei et al. demonstrated that the blue-eye-associated variants of rs12913832 and rs916977 at the *HERC2/OCA2* locus are associated with decreased time to first cSCC post-transplant in solid organ transplant recipients. In contrast, a protective role was found for several alleles located within genes influencing pigmentation, including the brown eye allele at *OCA1/HERC2* and the C allele for *SLC45A2*, a transporter protein that mediates melanin synthesis and contributes to skin pigmentation. OTRs homozygous for the brown eye alleles rs916977 (GG) and rs12913832 (AA) exhibited significant delays in the time to first cSCC post-transplant compared with individuals who were homozygous for the blue eye alleles, whereas the *SLC45A2* rs16891982 C allele was associated with a decreased HR for cSCC in univariate analysis [78]. These results could direct future recommendations in terms of screening guidelines and risk models for cSCC in OTRs.

#### 4.2.3. Differences in the Immune Landscape between ICPs and ISPs May Influence Responses to Targeted Therapy

The tumor immune microenvironment has gained interest in the recent literature. An understanding of the immune mechanisms involved in cSCC progression is necessary in order to guide therapeutic approaches. ISPs exhibit lower levels of CD8+ T-cells compared to their immunocompetent counterparts [74]. The role of cytotoxic T-cells in tumor progression has been well-established, with increased CD8+ T-cells being associated with good prognosis in many cancers [104,105,106,107,108,109]. It is therefore not surprising that the absence of an appropriate CD8+ T-cell response leads to poorer outcomes among ISPs. The increased expression of several other immune checkpoint receptors, including PD-L1, PD-L2, IDO, and TIM3, observed in metastatic cSCC among ICPs was also absent among ISPs [71]. This finding is of significance because immunotherapeutic agents targeting these receptors may be less useful among the immunosuppressed population.

## 5. Conclusions and Future Directions

We have provided an overview of the molecular alterations driving cSCC progression among immunocompetent and immunosuppressed populations based on the recent literature, providing a framework for future research.

Novel biomarkers in cSCC have emerged as promising therapeutic targets in this era of precision medicine and may alter the treatment paradigm of this deadly disease. Genes with roles in autophagy, perineural invasion, and metastasis could serve as new therapeutic targets for high-risk cSCC. MiRNA, circRNA, and lncRNA are novel regulators of gene expression that drive cSCC development at the transcriptomic level and present exciting therapeutic opportunities, which could be explored in future studies. Certain proteins demonstrate disease-progression-specific significance and an association with cSCC differentiation status and may act as molecular targets for selective tumor treatment. The complement system and the T-lymphocyte response influence cSCC progression by altering the tumor immune microenvironment. Immunotherapeutic agents targeting these signaling pathways could be investigated as therapeutic options.

Specific molecular alterations place ISPs at greater risk of cSCC development, and differences in tumor immune landscapes may explain the poorer therapeutic outcomes observed among ISPs compared to their immunocompetent counterparts. The unique molecular profile of cSCC in ISPs may be useful in predicting treatment responses, guiding the risk stratification process, and enhancing surveillance in this population.

## Figures and Tables

**Figure 1 cancers-15-01832-f001:**
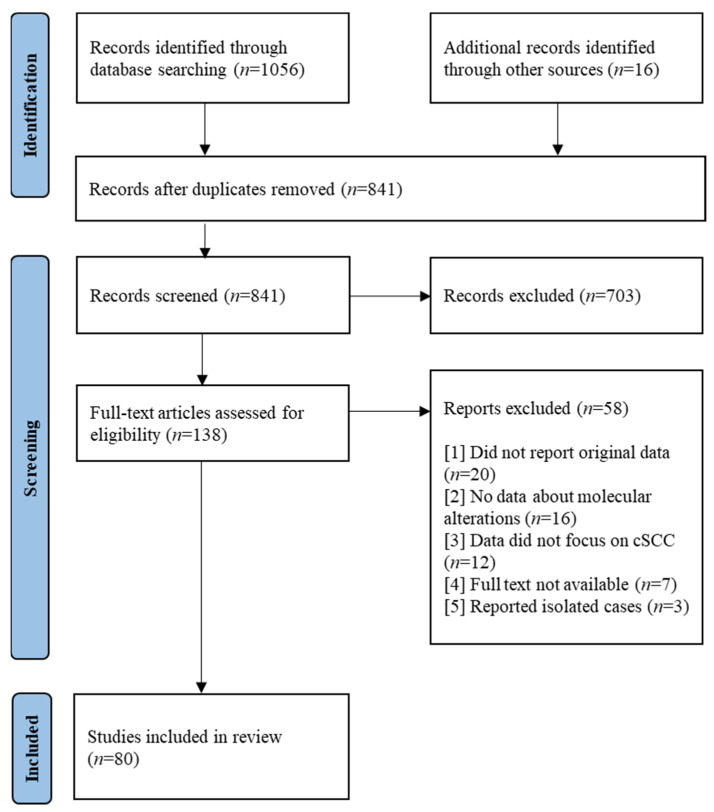
PRISMA 2020 flow diagram.

**Figure 2 cancers-15-01832-f002:**
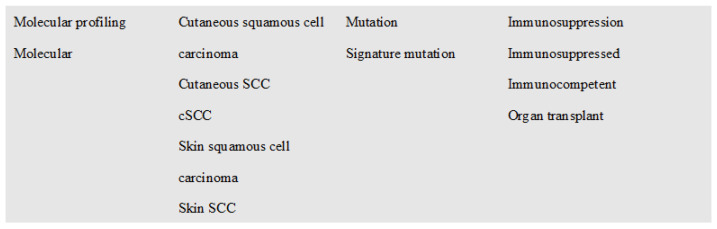
Representation of search terms used for the database search. Search terms within each box were connected by the Boolean operator OR. Each box represents a core search domain connected by the Boolean operator AND.

**Table 1 cancers-15-01832-t001:** Genes involved in perineural invasion.

Gene	Study Population	Results	Author	Year
*BCL2L1*	24 human cSCC samples (9 cSCCHN, 7 cSCCHN with incidental PNI, 8 cSCCHN with clinical PNI)	Downregulated in cSCCHN with PNI vs. without PNI (node degree of 36)	Zheng et al. [4]	2018
*ERBB2*		Upregulated in cSCCHN with PNI vs. without PNI (node degree of 31)		
*HIF1A*		Upregulated in cSCCHN with PNI vs. without PNI (node degree of 30)		
*MAPK8*		Upregulated in cSCCHN with PNI vs. without PNI (highest node degree of 41)		
*MTOR*		Downregulated in cSCCHN with PNI vs. without PNI (node degree of 34)		
*MYC*		Downregulated in cSCCHN with PNI vs. without PNI (node degree of 42)		
*PPARγ*		Downregulated in cSCCHN with PNI vs. without PNI (node degree of 32)		
*RAB23*		*RAB23* gene expression was positively correlated with *HIF1A* (*p* = 0.001, r = 0.690), *MAPK8* (*p* = 0.007, r = 0.583) and *ARFGAP1* (*p* = 0.000, r = 0.655), but negatively associated with *MTOR* (*p* = 0.002, r = −0.748) and *BCL2L1* (*p* = 0.015, r = −0.528)		
*TNF*		Downregulated in cSCCHN with PNI vs. without PNI (highest node degree of 44)		
*FGFR2*	10 cases of high-risk head and neck cSCC	Somatic missense mutations in *FGFR2* (40%) were exclusively seen in patients with PNI. Two novel mutations, *FGFR2 A380D* and *D528N* were observed in this cohort	C Zilberg et al. [6]	2018
*C3*	45 cases of human HNcSCC stratified into 3 groups (Extensive *n* = 25, Focal *n* = 11 and Non PNI *n* = 9)	Top 10 DEG identified between EXT PNI vs. FOC_NON (Padj < 0.01, log2FC = 3.237621)	Eviston et al. [5]	2021
*FERMT2*		Top 10 DEG identified between EXT PNI vs. FOC_NON (Padj < 0.01, log2FC = 1.717795)		
*HGF*		Top 10 DEG identified between EXT PNI vs. FOC_NON (Padj < 0.01, log2FC = 2.856216)		
*NR4A3*		Top 10 DEG identified between EXT PNI vs. FOC_NON (Padj < 0.01, log2FC = 2.830313)		
*PTGIS*		Top 10 DEG identified between EXT PNI vs. FOC_NON (Padj < 0.01, log2FC = 4.265134)		
*SAMSN1*		Top 10 DEG identified between EXT PNI vs. FOC_NON (Padj < 0.01, log2FC = 1.530389)		
*SRGN*		Top 10 DEG identified between EXT PNI vs. FOC_NON (Padj < 0.01, log2FC = 2.030587)		
*TIMP1*		Top 10 DEG identified between EXT PNI vs. FOC_NON (Padj < 0.01, log2FC = 2.198932)		
*THBS4*		Top 10 DEG identified between EXT PNI vs. FOC_NON (Padj < 0.01, log2FC = 4.996716)		
*VCAN*		Top 10 DEG identified between EXT PNI vs. FOC_NON (Padj < 0.01, log2FC = 1.849012)		

*ARFGAP1*, ADP ribosylation factor GTPase activating protein 1; *BCL2L1*, B-cell lymphoma 2 like 1; *C3*, complement 3; cSCCHN, head and neck cutaneous squamous cell carcinoma; *ERBB2*, erb-b2 receptor tyrosine kinase 2; EXT PNI, extensive perineural invasion; *FERMT2*, fermitin family homolog 2; *FGFR2*, fibroblast growth factor receptor 2; FOC_NON, combined focal and non PNI; *HGF*, hepatocyte growth factor; *HIF1A*, hypoxia-inducible factor 1α; log2FC, log2 fold change; *MAPK8*, mitogen-activated protein kinase 8; *MTOR*, mechanistic target of rapamycin kinase; *NR4A3*, nuclear receptor subfamily 4 group A member 3; Padj, adjusted *p*-value; *PPARγ*, peroxisome proliferator activated receptor γ; *PTGIS*, prostaglandin I2 synthase; *SRGN*, serglycin; *THBS4*, thrombospondin-4; *TIMP1*, tissue inhibitor matrix metalloproteinase 1; *TNF*, tumor necrosis factor; *VCAN*, versican.

**Table 2 cancers-15-01832-t002:** Genes involved in metastasis.

Gene	Study Population	Results	Author	Year
*KMT2D*	Human metastatic cSCC and primary non-metastatic cSCC	Higher rates of mutation in the metastatic samples (62%) relative to non-metastatic ones (31%)	Yilmaz et al. [8]	2017
*TP53*		Higher mutation frequencies in metastatic disease compared to localized disease (85% vs. 54% respectively; *p* < 0.0001)		
*TERT*	152 cSCC samples from 122 patients (in situ cSCC, *n* = 31; invasive cSCC, *n* = 121)	*TERTp* mutations were significantly more frequent in cases that recurred (13 out of 17 cases [76.5%] vs. 29 of 104 cases [27.9%] [*p* < 0.001]). TERTp mutation was identified as an independent predictor of recurrence (OR, 8.11; *p* = 0.002, multivariate analysis)	Campos et al. [10]	2018
*CDKN2A*	20 case-matched localized (10) and metastatic (10) high-risk cSCC	One of the most frequently mutated genes in localized (20%) and metastatic cSCC (40%)	M.B. Lobl et al. [9]	2020
*ERBB4*		Seen only in localized cSCC (20%). *ERBB4* and *STK11* were found to be significantly co-occurring in localized high-risk SCC (pair wise Fisher’s exact test *p* < 0.05)		
*HRAS*		Seen only in metastatic cSCC (20%)		
*KDR*		One of the most frequently mutated genes in localized (40%) and metastatic cSCC (30%)		
*KIT*		One of the most frequently mutated genes in localized (10%) and metastatic cSCC (20%)		
*NOTCH1*		One of the most frequently mutated genes in localized (20%) and metastatic cSCC (10%)		
*PTEN*		One of the most frequently mutated genes in localized (10%) and metastatic cSCC (20%)		
*SMAD4*		One of the most frequently mutated genes in localized (30%) and metastatic cSCC (20%)		
*STK11*		Seen only in localized cSCC (30%). *ERBB4* and *STK11* were found to be significantly co-occurring in localized high-risk SCC (pair wise Fisher’s exact test *p* < 0.05)		
*TP53*		One of the most frequently mutated genes in localized (70%) and metastatic cSCC (70%)		
*ITGA5*	cSCCHN from 50 patients. 21 PRI−, 14 PRI+, 15 MET, matched SES	Upregulated in MET vs. PRI+ (log2FC > 0.58, Padj < 0.05). Upregulated in MET vs. PRI- (log2FC > 0.58, Padj < 0.05)	Minaei et al. [7]	2022
*MMP1*		Upregulated in MET vs. PRI- (log2FC > 0.58, Padj < 0.05)		
*MMP10*		Shared upregulated gene in MET vs. SES and PRI+ vs. SES (log2FC > 1, Padj < 0.05)		
*MMP13*		Increased in MET vs. SES (log2FC > 1, Padj < 0.05), increased in PRI+ vs. SES		
*PLAU*		Shared upregulated gene in MET vs. SES and PRI+ vs. SES (log2FC > 1, Padj < 0.05). Upregulated in MET vs. PRI- (log2FC > 0.58, Padj < 0.05)		
*PLAUR*		Upregulated in MET vs. PRI- (log2FC > 0.58, Padj < 0.05)		
*TIMP1*		Upregulated in MET vs. PRI+ (log2FC > 0.58, Padj < 0.05)		
*TIMP4*		Downregulated in MET vs. SES (log2FC < 1, Padj < 0.05) and PRI+ vs. SES (log2FC < 1, Padj < 0.05)		
*VEGFA*		Upregulated in MET vs. PRI+ (log2FC > 0.58, Padj < 0.05). Upregulated in MET vs. PRI- (log2FC > 0.58, Padj < 0.05)		

*CDKN2A*, cyclin-dependent kinase inhibitor 2A; *ERBB 4*, Erb-B2 receptor tyrosine kinase 4; *HRAS*, Harvey rat sarcoma viral oncogene homolog; *ITGA5*, integrin subunit alpha 5; *KDR*, kinase insert domain receptor; *KMT2D*, lysine (K)-specific methyltransferase 2D; log2FC, log2 fold change; MET, patients with lymph node metastases but with no available primary tumor; *MMP*, matrix metalloproteinases; Padj, adjusted *p*-value; *PLAU*, urokinase plasminogen activator; *PLAUR*, urokinase plasminogen activator receptor; PRI−, locally confined tumors; PRI+, primary tumors that had metastasized; *PTEN*, phosphatase and tensin homolog; SES, sun-exposed skin; *STK11*, serine/threonine kinase 11; *TERTp*, telomerase reverse transcriptase promoter; *TIMP1*, tissue inhibitor of metalloproteinase-1; *TIMP4*, tissue inhibitor of metalloproteinase-4; *TP53*, tumor protein p53; *VEGFA*, vascular endothelial growth factor A.

**Table 3 cancers-15-01832-t003:** Genes with tumor-suppressive roles.

Gene	Study Population	Results	Author	Year
*RIPK4*	Mouse models	The onset of tumors commenced as early as 8 weeks in *RIPK4* cKO mice and 13 weeks in WT littermates. At week 11, 100% of cKO animals developed skin lesions, whereas more than 50% of WT animals remained tumor free even after 15 weeks	P Lee et al. [12]	2017
*HOXA9*	Human cSCC cell lines (A431, and HSC-1), control cell line (HaCaT)	*HOXA9* was downregulated in all cSCC cell lines compared with the primary keratinocytes and the HaCaT keratinocytes (*p* < 0.001). MiR-365 expression was inversely correlated with *HOXA9* expression in these cell lines	Zhou et al. [15]	2018
(Genes encoding) RNase H2	Mouse models	Loss of RNase H2 in the epidermis resulted in spontaneous DNA damage (increased numbers of repair foci, increased transcript levels of p53-inducible genes) and resulted in progression to skin cancer (at least KIN stage) in 100% of cases within the first year of life	Hiller et al. [13]	2018
*SFRP1*	A3886 (skin cutaneous SCC cell line), MCF-10A (control), and MDA-MB-231 (TNBC) cell lines	Sfrp1 −/− and Sfrp1+/− mice papilloma formation appears earlier by 3–4 weeks and 2–3 weeks, respectively, compared with WT mice	Sunkara et al. [14]	2020
*CREBBP*	Mouse models, human cSCC cell lines: A431(CRL–1555), SCC13, COLO16	Forced depletion of *CREBBP* increased cellular proliferation relative to control (1-factor ANOVA, *p* < 0.05). Knockdown of *CREBBP* expression led to larger and significantly more colonies relative to control (*p* = 0.018)	Aiderus et al. [17]	2021
*KMT2C*		Forced depletion of *KMT2C* increased cellular proliferation relative to control (1-factor ANOVA, *p* < 0.05). Knockdown of *KMT2C* accelerated in vitro proliferation and in vivo xenograft growth		
*CYLD*	Immunocompetent mice	Tumor multiplicity was higher in Control/TgAC mice vs. *K5-CYLDwt/TgAC* mice (*p* < 0.01). Lower levels of *NF-kB* activation found in *K5-CYLDwt/TgAC* tumors	Alameda et al. [11]	2021
*KANSL1*	Matched tumor and blood DNA from 25 patients with regional metastases of cSCCHN	*KANSL1* (Ch17q) showed deletion in 32% of tumor samples	Thind et al. [16]	2022
*PTPRD*		Deletion of *PTPRD* (Chr9p) was observed in 20% of tumor samples		

cKO, complete knockout; *CREBBP*, CREB binding protein; cSCCHN, cSCC of the head and neck; *HOXA9*, homeobox A9; *K5-CYLDwt*, transgenic mice that expressed the wild-type form of *CYLD* under the control of the keratin 5 (K5) promoter; *KANSL1*, KAT8 regulatory NSL complex subunit 1; KIN, keratinocyte intraepithelial neoplasia; *KMTC2*, lysine (K)-specific methyltransferase 2C; *NF-kB*, nuclear factor-κB; NHEK, normal human epidermal keratinocytes; *PTPRD*, protein-tyrosine phosphatase delta; *RIPK4*, receptor interacting protein 4; RNase H2, ribonuclease H2; *SFRP1*, secreted frizzled-related protein 1; TgAC, animals previously bred with TgAC, which carried an activated Ha-ras trans-gene that triggers the classic tumor initiation event; TNBC, triple-negative breast cancer; WT, wild-type.

**Table 5 cancers-15-01832-t005:** Epigenetic alterations in cSCC.

Epigenetic Biomarker	Study Population	Results	Author	Year
*bHLH TFs*	Genetically engineered mouse models: Lgr5CreER and K14CreER mice	Upregulated during EMT and enriched in the open chromatin regions of TMCs in 20–45% of targets	Latil et al. [26]	2017
*Ets1*		Upregulated during EMT and positively associated with gene expression in 10% of targets. Upregulated during tumorigenesis in 37% of targets (*p* = 1 × 10^−10^)		
*Jun/AP1*		Upregulated during EMT and enriched in the open chromatin regions of TMCs in 42% of targets. Enriched in the open chromatin regions during tumorigenesis in 65% of targets (*p* = 1 × 10^−130^)		
*NF1*		Upregulated during EMT and enriched in the open chromatin regions of TMCs in 45% of targets		
*Nf-kb*		Enriched in the open chromatin regions during tumorigenesis in 22% of targets (*p* = 1 × 10^−10^)		
*Nfatc*		Upregulated during EMT and enriched in the open chromatin regions of TMCs in 27% of targets		
*Runx*		Enriched in the open chromatin regions during tumorigenesis in 29% of targets (*p* = 1 × 10^−10^)		
*Smad2*		Upregulated during EMT and enriched in the open chromatin regions of TMCs in 37% of targets		
*TEAD*		Enriched in the open chromatin regions during tumorigenesis in 25% of targets (*p* = 1 × 10^−8^)		
*DNA methylation*	23 human cSCC samples diagnosed at the following stages: AK, early invasive carcinoma, high-risk non-metastatic carcinoma and high-risk carcinoma with nodal metastasis	Initial invasive group showed lower methylation levels than premalignant actinic keratosis. Hypermethylation of all substructures in both high-risk non-metastatic and metastatic groups compared to low-risk initial invasive cSCC samples (*p* < 0.001, two-sided *t*-test)	Hervás-Marín et al. [28]	2019
*ID4*	8 pairs of matched human cSCC and adjacent normal skin tissues; sun-exposed normal skin in the head and neck region (*n* = 60) from normal patients and distal non-exposed normal skin from cSCC patients (*n* = 60)	*ID4* expression was downregulated in cSCC (*p* = 0.0111) and correlated with increased levels of promoter methylation (*p* = 0.00295)	L. Li et al. [29]	2020
*UCHL1*		*UCHL1* expression was downregulated in cSCC (*p* = 0.0205) and correlated with increased levels of promoter methylation (*p* = 0.0499)		
*ACTL6A*	SCC-13 and HaCaT cell lines	*ACTL6A* knockdown reduces SCC cell proliferation, spheroid formation, invasion and migration	Shrestha et al. [27]	2020
*Filip1l*	Mouse cSCC, human cSCC, NHEK	In murine cSCC tumours, the Filip1l protein levels were reduced compared to matched controls (paired t-test, *p* = 0.0026). In human cSCC, *FILIP1L* protein levels were increased in one cSCC cell line, similar to NHK in 4/12 cSCC cell lines and lower (i.e., below 2/3 of NHK means) than NHEK in 7/12 cSCC cell lines	K. Roth et al. [30]	2021
*ZMIZ1*	Mouse models, human cSCC cell lines: A431(CRL–1555), SCC13, COLO16	All cSCC tumors with *Zmiz1/2* insertions had inactivating insertions in at least one gene involved in chromatin remodeling	Aiderus et al. [17]	2021
*ZMIZ2*				

*ACTL6A*, actin-like 6A; *bHLH* TFs, basic helix loop helix transcription factors; EMT, epithelial–mesenchymal transition; *FILIP1L*, filamin A interacting protein 1-like; *ID4*, inhibitor of DNA binding 4; *NF-kB*, nuclear factor-κB; *NFATC1*, nuclear factor of activated T-cells, cytoplasmic 1; *Runx*, runt-related transcription factor; *TEAD*, TEA domain family member; TMCs, tumor mesenchymal-like cells; *UCHL1*, ubiquitin carboxyl-terminal esterase L1.

**Table 7 cancers-15-01832-t007:** The immune microenvironment of cSCC in the immunosuppressed.

Immune Biomarker	Study Population	Results	Author	Year
CGRP	cSCC from 34 OTRs; pain-associated (*n* = 18), without pain (*n* = 16)	No difference in CGRP expression levels in cSCC with pain vs. cSCC without pain in OTRs.	Frauenfelder et al. [76]	2017
NGF		No difference in NGF expression levels in cSCC with pain vs. cSCC without pain in OTRs.		
IL-1β		No difference in IL-1β expression levels in cSCC with pain vs. cSCC without pain in OTRs.		
PGE2		cSCC with pain is associated with increased levels of PGE_2_ compared with cSCC without pain (OR = 1.9, 95% CI = 1.1–3.4, *p* = 0.002), adjusted for age and sex.		
POMC		cSCC with pain was associated with increased levels of POMC compared with cSCC without pain (OR = 1.5, 95% CI = 0.99–2.0, *p* = 0.05), adjusted for age and sex.		
TNF-α		cSCC with pain was associated with increased levels of TNF-α compared with cSCC without pain (adjusted OR = 1.4, 95% CI = 0.99–2.0, *p* = 0.05).		
AIM2	Primary (*n* = 5) and metastatic (*n* = 3) human cSCC cell lines, NHEK (*n* = 5). Tissue samples: normal sun-protected skin (*n* = 15), AK (*n* = 71), cSCCIS (*n* = 60), and UV-induced cSCC (*n* = 81)	In OTR derived tissues, AIM2 expression was significantly more abundant in cSCC (*n* = 57) compared with cSCCIS (*n* = 59, *p* < 0.001)	Farshchian et al. [65]	2017
B7-H3	SCC from 42 ICP and 24 ISP (13 OTRs, 8 HIV, and 3 others)	Tumor expression of B7-H3 was higher in in ICP vs. ISP (Median 60 vs. 28%, *p* = + 0.025)	Varki et al. [75]	2018
PD-L1		No difference in PD-L1 expression between ICP and ISP (*p* = 0.5).		
CD8+ T-cells	cSCC (*n* = 5), TSCC (*n* = 6)	OTRs generally exhibited lower levels of CD8+ TILs (*n* = 6880 ICP; *n* = 2484 ISP, *p* < 0.05)	Frazzette et al. [74]	2020
IL-17A	14 human cSCC: ISP (*n* = 3), ICP (*n* = 11)	CD3^−^IL-17^+^ cells consist of over 90% of the total IL-17A-producing cells in the tumor tissue from ISP, while the ratio of CD3^+^IL-17^+^ versus CD3^−^IL-17^+^ cells vary in ICP	Sun et al. [73]	2020
B cells	Human cSCC from 31 patients: 9 NP, 22 DP. Within the. DP group: 5 ISP and 17 ACIS	In ACIS patients there was significant increase in total B cell numbers and proliferating B cells in metastases compared to primary DP tumours. This increase was not evident among ISP	Ferguson et al. [71]	2022
IDO		The increased expression of IDO on all T cells in metastatic cSCC among ACIS patients was absent among ISP		
PD-L1		The increased expression of PD-L1 on CD8+ T cells in metastatic cSCC among ACIS patients was absent among ISP		
PD-L2		The increased expression of PD-L2 on CD4+ T cells in metastatic cSCC among ACIS patients was absent among ISP		
TIM3		TIM3 was increased on non-classical monocytes in metastatic tumours of ACIS patients compared to immunosuppressed patients		

AK, actinic keratosis; ACIS, absence of clinical immune suppression; B7-H3, B7 homolog 3 protein; CGRP, calcitonin gene-related peptide; CI, confidence interval; cSCCIS, cSCC in situ; DP, progression to metastases; IDO, indoleamine 2,3-dioxygenase; IL, interleukin; NGF, nerve growth factor; NHEK, normal human epidermal keratinocyte; NP, non-progressors; OTR, organ transplant recipient; OR, odds ratio; PD-L1, programmed death-ligand 1; PD-L2, programmed death-ligand 2; PGE2, prostaglandin E2; POMC, proopiomelanocortin; siRNA, small interfering RNA; TIM3, T-cell immunoglobulin and mucin domain 3; TNF-α, tumor necrosis factor-α; TSCC, transplant-associated SCC.

**Table 8 cancers-15-01832-t008:** Genetic polymorphisms associated with cSCC in the immunosuppressed.

Genetic Polymorphism		Study Population	Results	Author	Year
*HERC2*	rs916977, rs12913832 brown eye allele compared with blue eye allele	386 OTRs with cSCC and without	OTRs homozygous for brown eye alleles rs916977 (GG) and rs12913832 (AA) had significant delays of time to first cSCC after transplant vs. OTRs homozygous for blue eye alleles (HR = 0.34, *p* < 0.001; HR = 0.54, *p* = 0.012, respectively).	Wei et al. [78]	2017
*OCA2*					
*IRF4*	rs12203592 T allele	388 OTRs with cSCC and without	The *IRF4* rs12203592 T allele was associated with a significantly increased hazard for time to first cSCC (HR = 1.36, *p* = 0.02, univariate analysis). This association was maintained when adjusted for age, gender, organ transplanted, and Fitzpatrick skin type (HR = 1.34, *p* = 0.04).	Asgari et al. [77]	2017
*SLC45A2*	rs16891982 C allele		The *SLC45A2* rs16891982 C allele was associated with a decreased hazard for cSCC (HR = 0.58, *p* = 0.04, univariate analysis); this effect was comparable but not significant using the multivariate model (HR = 0.74, *p* = 0.06).		
Upstream of *RP1163E5.6*, *FBXO25*, and *OR4F2*	rs34567942	61 OTRs with cSCC and 908 OTRs without cSCC	GWAS identified one SNV, rs34567942, to be significantly associated with cSCC in OTRs (*p*-value threshold of 5 × 10^−8^)	Kuzmanov et al. [79]	2019

*HERC2*, HECT and RLD domain-containing E3 ubiquitin protein ligase 2; HR, hazard ratio; *IRF4*, interferon regulatory factor 4; *OCA2*, oculocutaneous albinism 2; OTRs, organ transplant recipients; SNV, single nucleotide variant.

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
