# Peer review of "Molecular Alterations in Cutaneous Squamous Cell Carcinoma in Immunocompetent and Immunosuppressed Hosts—A Systematic Review"

_cancers, 2023, doi:10.3390/cancers15061832_

Round 1

Reviewer 1 Report

This is a very interesting and worthwhile systematic review discussing the molecular profile of cutaneous squamous cell carcinoma (cSCC) among immunocompetent patients (ICP) and immunosuppressed patients (ISP) at the genetic, epigenetic, transcriptomic and proteometabolomic levels. The review also describes key differences in the tumor immune microenvironment between ICP and ISP populations. Based on the literature analyses, the authors suggested that the unique molecular profile of cSCC in ISP may be useful in predicting treatment response, guiding risk stratification, and enhancing surveillance in this population.

My comments are minor:

1.      The terms ‘immunocompetent’ and ‘non-immunosuppressed’ have been interchangeably used though out the manuscript. I would suggest using any one of them mainly to simplify for the readers.

2.      The starting paragraph for several subsections appears to be abrupt. The authors are suggested to use 1-2 introductory sentences before discussing the findings of the systematic review. For example, in line 70, it would be ideal to first introduce the term ‘perineural invasion’, and its incidences, frequencies, or any other related relevant information of cSCC.

3.      Several pieces of information appear to be repetitive in Tables. For example, in Table 1, a study by Eviston et al (2021) has been presented in 10 separate columns where all the information (except the name of the genes) is the same. It is also not clear how Tables were organized. Is it based on fold-change or up/downregulated genes? Similar attention will be needed for other Tables too.

4.      Some typos/presentation problems were noticed. E.g. ‘polymorphysms’ (line 638); ‘Our review identified…’(line 399. This should be reworded. It is a review of existing information); etc. Another round of careful editing to make these all consistent will clean up the manuscript.

5.      What is meant by signature 32 (line 495)? Please elaborate.

6.      The authors are suggested to include in the ‘Conclusion and future directions’ section how this systematic review could potentially inform future studies and clinical practices.

Reviewer 2 Report

Abstract: Methodology, results, and conclusions need improvement. 

Introduction: The general data on cSCC need improvement (local occurrence, incidence, mortality, etc.). 

Materials and methods: 

The authors stated: 'The PRISMA-P (Preferred Report Items for Systematic Review and Meta-Analysis) guideline guided the methodology of this systematic review (Figure 1)'. Where is the meta-analysis data?

Why did the authors choose the period 2017 to 2022? 

Why was the search not limited by the design of the studies (retrospective, prospective, cohort, clinical series of cases, or clinical case)? How did the authors assess the quality of the studies?

The specific criteria used to exclude are unclear.

Conclusion: The results of the different articles are not summarised in the conclusions.
